# MM-UNet: Meta Mamba UNet for Volumetric Medical Image Segmentation

## Abstract

State Space Models (SSMs) have recently demonstrated outstanding performance in long-sequence modeling, particularly in natural language processing. However, their direct application to medical image segmentation poses several challenges. SSMs, originally designed for 1D sequences, struggle with 3D spatial structures in medical images due to discontinuities introduced by flattening. Additionally, SSMs have difficulty fitting high-variance data, which is common in medical imaging. In this paper, we analyze the intrinsic limitations of SSMs in medical image segmentation and propose a unified U-shaped encoder-decoder architecture, Meta Mamba UNet (MM-UNet), designed to leverage the advantages of SSMs while mitigating their drawbacks. MM-UNet incorporates hybrid modules that integrate SSMs within residual connections, reducing variance and improving performance. Furthermore, we introduce a novel bi-directional scan order strategy to alleviate discontinuities when processing medical images. Extensive experiments on AMOS22 Ji et al. (2022) and Synapse Landman et al. (2015) datasets demonstrate the superiority of MM-UNet over state-of-the-art methods. MM-UNet achieves a Dice score of 91.0% on AMOS22, surpassing nnUNet by 1.7%, and a Dice score of 87.1% on Synapse. These results confirm the effectiveness of integrating SSMs in medical image segmentation through architectural design optimizations.

## 1 Introduction

Medical image segmentation plays a crucial role in biomedical image analysis, aiding in disease diagnosis, abnormality detection, and surgical planning. In recent years, significant progress has been achieved in medical image segmentation through deep learning-based methods Ronneberger et al. (2015); Isensee et al. (2019); Hatamizadeh et al. (2022); Zhou et al. (2021), with convolutional neural networks (CNNs) and vision transformers (ViTs) emerging as dominant architectures. However, CNNs are inherently limited by their local receptive fields, making them less effective at capturing long-range dependencies. ViTs suffer from high computational complexity due to their quadratic attention mechanism, which restricts their ability to efficiently model global dependencies.

Recently, State Space Models (SSMs) have gained significant attention in natural language processing (NLP) due to their ability to model long sequences with linear time complexity. Their success has also influenced the field of computer vision. This raises the challenge of determining whether SSMs, originally designed for 1D sequences, can achieve high performance when applied to images with at least two spatial dimensions, particularly medical images with 3D spatial information.

To investigate this, we first explore the intrinsic limitations of SSMs in image-based tasks. Typically, SSMs process images by flattening them in a predefined order, such as width-first (H-W order). We conduct experiments using the Structured State Space Sequence Model (S4) Gu et al. (2021) to fit 1D sequences derived from 2D medical images (see Fig. 2 (a)). The results indicate that S4 struggles to fit points that deviate significantly from the mean, particularly those with high variance. Additionally, flattening images into a 1D sequence introduces discontinuities, making it difficult for SSMs to infer relationships across adjacent rows. However, these discontinuities can be mitigated using an inverse scan order, which enhances the ability of SSMs to model spatial structures more effectively.

Building on the advancements of Transformers in medical image segmentation (e.g., Swin-UNETR Tang et al. (2022), nnFormer Zhou et al. (2021)), several U-shaped architectures incorporating Mamba have been developed. These models take inspiration from Transformer-based

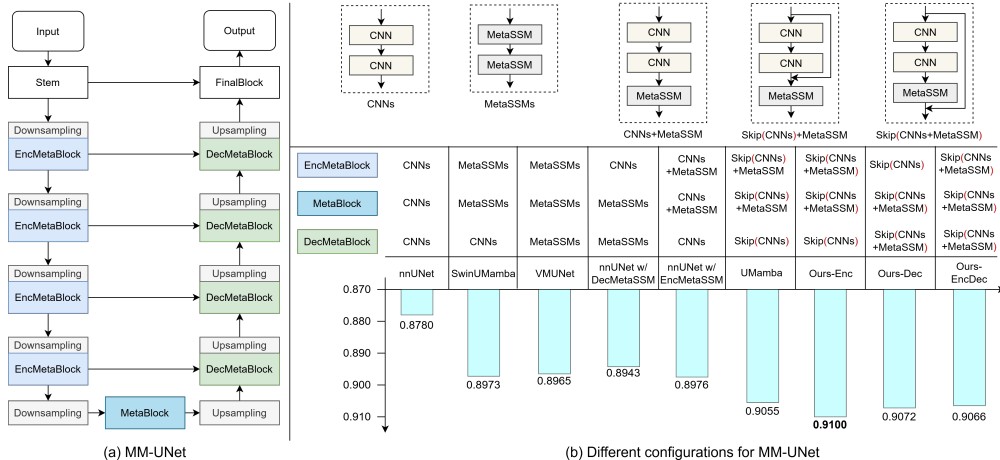

Figure 1: (a) Overview of our proposed MM-UNet architecture. (b) Experiments replacing meta blocks in MM-UNet with different modules, including pure CNN-based, hybrid, and pure SSM-based modules. Skip connections represent residual connections.

segmentation approaches, including SegMamba, which employs a pure Mamba encoder with a CNN-based decoder, VM-UNet, a fully Mamba-based U-Net, and U-Mamba, a hybrid model combining Mamba and CNN. A key consideration is determining the most effective way to integrate Mamba into a U-shaped architecture to maximize performance. The comparison between pure Mamba-based models and hybrid approaches remains crucial in identifying the optimal design for medical segmentation.

To address this, we introduce MM-UNet (Meta Mamba UNet), a unified U-shaped encoder-decoder architecture with skip connections, as shown in Fig. 1. Each stage of the encoder, bottleneck, and decoder consists of a replaceable meta-block, allowing flexibility in module selection. Five different configurations are evaluated, including pure CNN-based, hybrid CNN-Mamba, and pure Mamba-based modules. Extensive experiments reveal that a hybrid module, where SSMs are placed after two sequential CNNs and within residual connections, delivers the best performance. Additionally, replacing meta-blocks in the encoder and bottleneck with this hybrid module further enhances results.

These improvements are attributed to the ability of pre-trained feature maps behind CNN layers and inside residual connections to exhibit lower variance compared to those outside residual connections (Fig. 2 (b)), reinforcing the observation that SSMs struggle with high-variance inputs. With the macro-architecture defined, optimal scan orders for SSMs are explored. Fig. 3 illustrates various scan order strategies, and results indicate that a simple bi-directional scan order achieves the best performance. This strategy is adopted in the final model to maximize effectiveness.

In summary, this paper presents the following key contributions:

- We propose MM-UNet, a unified U-shaped encoder-decoder architecture with skip connections, capable of representing existing Mamba-based segmentation models.
- We identify the challenges of SSMs in handling high-variance data and discontinuities caused by flattening spatial dimensions, and address these limitations through architectural refinements.
- We design a hybrid module that integrates SSMs within residual connections, reducing variance and improving segmentation accuracy.
- We introduce a bi-directional scan order strategy to mitigate discontinuities and enhance spatial coherence in medical image segmentation.
- We conduct extensive experiments on AMOS2022 Ji et al. (2022) and Synapse Landman et al. (2015), demonstrating that MM-UNet achieves state-of-the-art performance with a 91.0% Dice score on AMOS2022, surpassing nnUNet by 1.7%, and an 87.1% Dice score on Synapse.

## 2 RELATED WORK

### 2.1 MEDICAL IMAGE SEGMENTATION

Convolutional Neural Networks (CNNs), particularly the encoder-decoder-based U-Net Ronneberger et al. (2015) and its variants Zhou et al. (2018); Çiçek et al. (2016); Milletari et al. (2016), have demon-

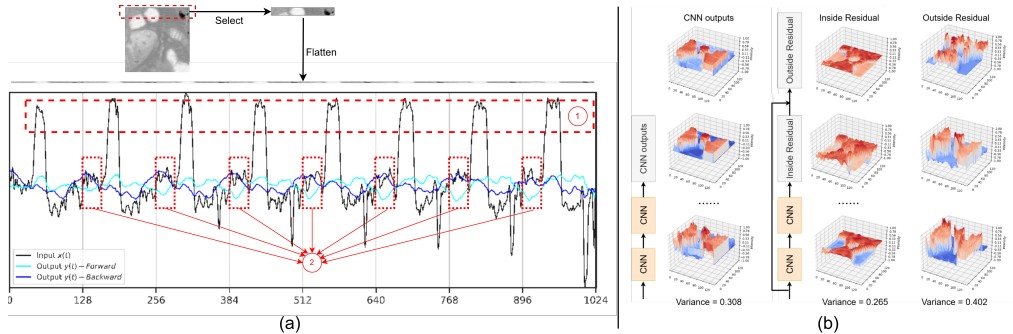

Figure 2: (a) Experiments using S4 to fit flattened 2D medical images. (b) The intensity distribution of feature maps inside and outside a residual connection from a pre-trained model, as well as from two sequential convolutional layers.

strated strong performance and play a crucial role in medical image segmentation. nnUNet Isensee et al. (2019) is a self-adapting framework for U-Net-based segmentation that incorporates automated preprocessing, dataset attribute analysis, optimized training strategies, and postprocessing techniques. Due to its robustness and strong performance across various segmentation tasks, nnUNet has been widely adopted by researchers in medical image analysis challenges.

With the rise of Transformer-based models, several works Cao et al. (2021); Hatamizadeh et al. (2022); Tang et al. (2022); Zhou et al. (2021) have leveraged the powerful capabilities of Transformers for medical image segmentation, achieving remarkable progress. However, CNNs are inherently constrained by their local receptive fields, limiting their ability to model long-range dependencies. Meanwhile, Vision Transformers (ViTs) suffer from the high computational complexity of their self-attention mechanism, which only captures long-range dependencies within predefined windows, restricting global feature aggregation.

### 2.2 SSMs for Medical Image Segmentation

Recently, State Space Models (SSMs) Kalman (1960); Gu et al. (2021); Gu & Dao (2023) have demonstrated outstanding performance in the NLP domain, gaining significant attention due to their ability to capture long-range dependencies with linear time complexity. Their success has extended to the vision domain, where several studies Zhu et al. (2024); Liu et al. (2024b); Huang et al. (2024) have explored the integration of SSMs into computer vision tasks, achieving promising results.

SSMs have also been applied to medical image segmentation, where multiple studies Ma et al. (2024); Ruan & Xiang (2024); Xing et al. (2024) have incorporated SSMs into U-shaped architectures, resulting in both pure Mamba-based and hybrid models that perform well on several segmentation benchmarks. U-Mamba Ma et al. (2024) was the first U-shaped Mamba-based model, introducing Vision Mamba into the segmentation pipeline. VM-UNet Ruan & Xiang (2024) extends this idea by integrating bi-directional SSMs into the U-Net framework, creating a purely Mamba-based segmentation model. Inspired by SwinUNETR Hatamizadeh et al. (2021), Swin-UMamba employs a Mamba-based encoder with a pre-trained model for medical image segmentation.

Despite these advancements, most studies have focused on directly integrating SSMs into existing segmentation architectures without thoroughly investigating their intrinsic limitations in processing medical images. The application of SSMs to medical image segmentation presents unique challenges, including handling spatial discontinuities when flattening multi-dimensional images and managing high-variance medical image data. In this paper, we analyze these challenges and propose a unified Mamba-based UNet that effectively integrates SSMs while addressing their fundamental drawbacks, leading to improved segmentation performance.

## 3 The Proposed Method

### 3.1 State Space Sequence Models

State Space Sequence Models (SSMs) Gu (2023) originate from classical state space models, which map a one-dimensional input signal $x(t) \in \mathbb{R}$ to a one-dimensional output signal $y(t) \in \mathbb{R}$ via an

implicit N-dimensional latent state $h(t) \in \mathbb{R}^N$:

$$
\begin{aligned}
h'(t) &= \mathbf{A}h(t) + \mathbf{B}x(t), \\
y(t) &= \mathbf{C}h(t),
\end{aligned}
\tag{1}
$$

where the state matrix $\mathbf{A} \in \mathbb{R}^{N \times N}$, input transformation matrix $\mathbf{B} \in \mathbb{R}^{N \times 1}$, and output transformation matrix $\mathbf{C} \in \mathbb{R}^{1 \times N}$ are learnable parameters. SSMs offer linear computational complexity per time and support parallelized computation, making them efficient for long-sequence modeling.

To process discrete input sequences $\mathbf{x} = (x_0, x_1, ...) \in \mathbb{R}^L$, Structured State Space Sequence Models (S4) Gu et al. (2021) discretize the parameters in Eq. 1 using a step size $\Delta$, which defines the resolution of the continuous input $x(t)$. Specifically, the continuous parameters $\mathbf{A}, \mathbf{B}$ are converted into discrete parameters $\overline{\mathbf{A}}, \overline{\mathbf{B}}$ using the zero-order hold (ZOH) method, as follows:

$$
\begin{aligned}
\overline{\mathbf{A}} &= \exp(\Delta\mathbf{A}), \\
\overline{\mathbf{B}} &= (\Delta\mathbf{A})^{-1}(\exp(\Delta\mathbf{A}) - \mathbf{I}) \cdot \Delta\mathbf{B}.
\end{aligned}
\tag{2}
$$

After discretization, Eq. 1 is reformulated:

$$
\begin{aligned}
h_t &= \overline{\mathbf{A}}h_{t-1} + \overline{\mathbf{B}}x_t, \\
y_t &= \mathbf{C}h_t.
\end{aligned}
\tag{3}
$$

To enhance computational efficiency and scalability, the iterative process in Eq. 3 can be reformulated as a global convolution:

$$
\begin{aligned}
\overline{\mathbf{K}} &= (\mathbf{C}\overline{\mathbf{B}}, \mathbf{C}\overline{\mathbf{A}}\overline{\mathbf{B}}, ..., \mathbf{C}\overline{\mathbf{A}}^{L-1}\overline{\mathbf{B}}), \\
\mathbf{y} &= \mathbf{x} * \overline{\mathbf{K}},
\end{aligned}
\tag{4}
$$

where $L$ is the length of the input sequence $\mathbf{x}$, and $\overline{\mathbf{K}} \in \mathbb{R}^L$ represents the SSM convolution kernel.

S4 employs structured forms on the state matrix $\mathbf{A}$ and uses the High-Order Polynomial Projection Operator (HIPPO) Gu et al. (2020) for initialization, enabling deep sequence models with enhanced capabilities for efficient long-range reasoning. As an advanced sequence modeling approach, S4 has outperformed Transformers Vaswani et al. (2017) on the challenging Long Range Arena Benchmark Tay et al. (2020).

Recently, Mamba Gu & Dao (2023) has introduced significant advancements in SSM-based sequence modeling. Unlike traditional SSMs, Mamba incorporates an input-dependent selection mechanism, enabling more efficient information filtering. Moreover, it employs a hardware-aware algorithm that scales linearly with sequence length, allowing efficient recurrent computation via a scanning operation. Mamba has demonstrated superior computational speed compared to previous methods on modern hardware and has achieved state-of-the-art performance in various long-sequence modeling domains.

### 3.2 APPLYING MAMBA TO MEDICAL IMAGING

This section examines the application of State Space Models (SSMs) in medical imaging, focusing on their fundamental characteristics and limitations. SSMs are primarily designed for one-dimensional sequences, whereas images, including medical images, have multiple spatial dimensions.

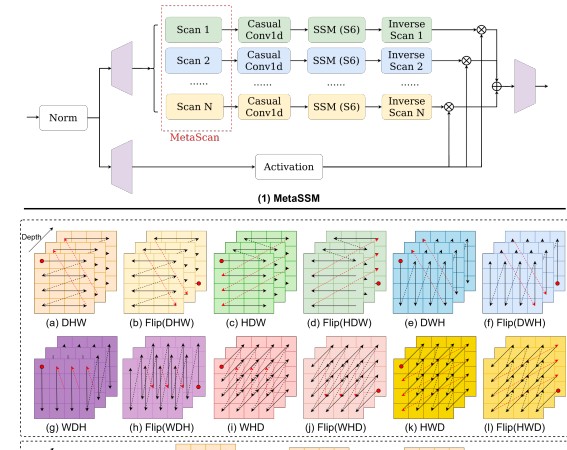

Figure 3: (1) Overview of MetaSSM architecture. (2) Evaluation of MetaScan scan orders.

| | Method | DSC ↑ |
|---|---|---|
| B1 | (a) DHW | 0.902 |
| B2 | (a) DHW + (b) flip(DHW) | **0.910** |
| B3 | (a) DHW + (b) flip(DHW) + (k) HWD | 0.898 |
| B4 | (a) + (b) + (c) + (d) + (e) + (f) | 0.908 |
| B5 | (a) + (b) + (c) + (d) + (e) + (f) + (g) + (h) + (i) + (j) + (k) + (l) | 0.907 |
| B6 | (m) 2D Scan w/ DHW + flip(DHW) | 0.900 |
| B7 | (n) 3D Window Scan w/ DHW + flip(DHW) | 0.902 |
| B8 | (o) Zigzag Scan w/ DHW + flip(DHW) | 0.909 |
| B9 | (p) Inclined Scan w/ DHW + flip(DHW) | 0.901 |

Table 1: Performance comparison of different MetaScan configurations with various scan orders.

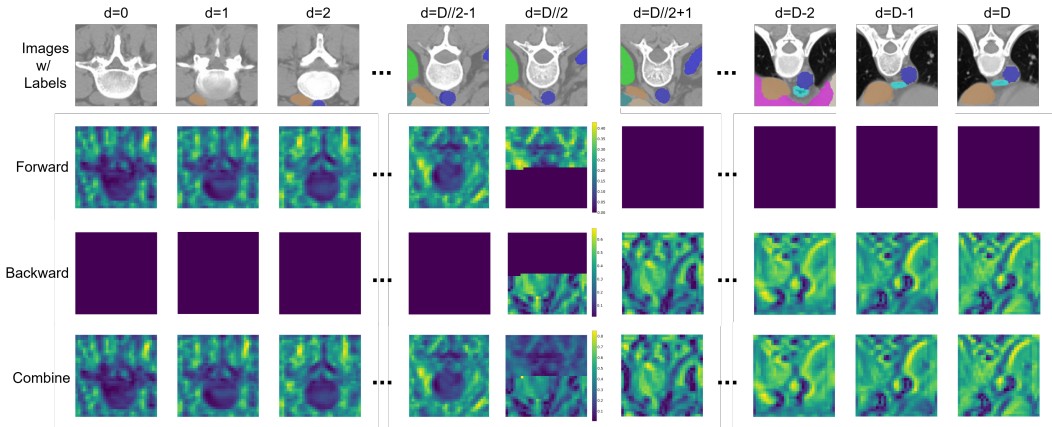

Figure 4: Visualization of attention maps of $QK^T$ for Mamba. Each attention map effectively captures image patterns across the temporal dimension, even when the 3D medical images are flattened into a 1D sequence as input for the MetaSSM blocks, highlighting the motivation for using SSMs in medical image segmentation.

To process images with SSMs, researchers typically flatten them using a predefined scan order, such as scanning row by row along the width dimension before the height dimension.

To analyze the impact of scan order on SSM performance, an experiment was conducted where a two-dimensional medical image of size $128 \times 128$ was flattened using a width-first order. The Structured State Space Sequence Model (S4) Gu et al. (2021) was then applied to fit the resulting one-dimensional sequence. For clarity, only the first eight rows of the width dimension are shown in Fig. 2 (a).

We observed that the S4 model struggles to fit points that deviate significantly from the mean. This issue arises due to S4's use of the High-Order Polynomial Projection Operator

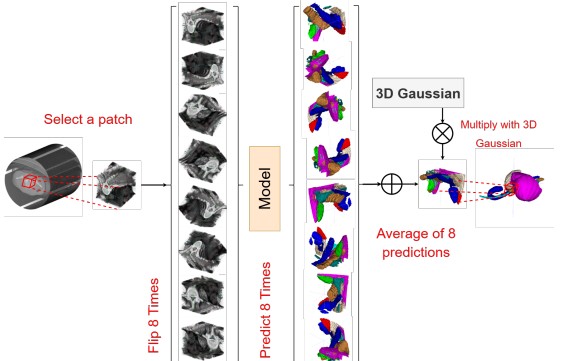

Figure 5: Flipping along axial, coronal, and sagittal dimensions during inference to improve performance. This operation is similar to applying multiple scan orders within Mamba blocks.

(HiPPO) Gu et al. (2020), which determines the coefficients of several polynomials that are combined to approximate the one-dimensional sequence, as shown in the case (1) of Fig. 2 (a). This process inherently smooths the predictions, which is problematic in medical imaging since different organs and tissues often exhibit high-intensity variations that are critical for accurate segmentation. To address this, reducing the variance of the input data is necessary when using SSMs.

Another challenge arises at the transition from the end of one row to the beginning of the next, indicated by the cyan line in the case (2) of Fig. 2 (a). In the red boxes, the model's predictions are nearly the inverse of the ground truth (black line). This discrepancy occurs because the beginning of a new row does not necessarily correlate with the end of the previous row, making it difficult for SSMs to infer relationships across these discontinuities. To mitigate this issue, discontinuities introduced by flattening images into one-dimensional sequences must be minimized.

Incorrect predictions caused by discontinuities in a particular scan order appear only within that order. A straightforward yet effective solution is to apply SSMs in the reverse scan order, where the beginning of a row in the original order becomes the end of a row in the reverse order. The blue line in Fig. 2 (a) illustrates this inverse ordering. The predictions in the red boxes improve significantly in this arrangement, demonstrating that using a pair of opposite-direction scan orders helps mitigate errors introduced by discontinuities.

Since images have multiple spatial dimensions, incorporating additional pairs of opposite-direction scan orders may seem beneficial. However, our experiments indicate that a single pair of opposite-direction scan orders can sufficiently represent the input images. Fig. 4 illustrates the attention maps for Mamba blocks, demonstrating that each attention map effectively captures image patterns.

Further, in our approach, during inference illustrated in Fig. 5, each image patch undergoes eight flips across axial, coronal, and sagittal dimensions, followed by averaging the predictions to improve performance. This operation functions similarly to applying multiple scan orders within Mamba blocks. Meanwhile, a 3D Gaussian will be multiplied by the averaged prediction to enhance the weight of the central region and weaken the weight of the peripheral region. The approach can mitigate the prediction errors caused by boundary discontinuities.

### 3.3 ARCHITECTURAL DESIGN OF MM-UNET

Recent advancements in medical image segmentation, driven by Transformer-based models such as SwinUNETR and nnFormer, have inspired the development of several U-shaped architectures that integrate Mamba modules. Examples include SegMamba, which utilizes a pure Mamba encoder combined with a CNN-based decoder; VM-UNet, which employs a fully Mamba-based U-Net; and U-Mamba, a hybrid approach combining Mamba with CNN components. Despite these developments, it remains unclear how best to integrate Mamba modules into U-shaped architectures to optimize segmentation performance, and whether a hybrid model could outperform purely Mamba-based approaches.

To address this challenge, we propose MM-UNet, a unified meta U-Net architecture illustrated in Fig. 1(a). MM-UNet features a symmetric encoder-decoder structure with skip connections, facilitating the integration of low-level fine-grained details and high-level semantic features. The model is composed of a stem module for channel expansion, an encoder that stacks several stages with each stage consisting of a downsampling layer followed by encoder meta blocks (EncMetaBlock), a bottleneck module (MetaBlock), and a decoder that mirrors the encoder structure with corresponding upsampling layers and decoder meta blocks (DecMetaBlock). Finally, a FinalBlock adjusts the channel dimensions to produce the segmentation output.

The EncMetaBlock, DecMetaBlock, and MetaBlock modules are designed to be interchangeable, enabling various configurations including CNN-based modules, pure Mamba-based modules, and hybrid CNN-Mamba modules, as depicted in Fig. 1(b). For instance, replacing all three blocks with sequential convolutional layers results in an nnUNet-like architecture, while employing purely sequential Mamba blocks results in a VM-UNet-like architecture.

We systematically evaluate different configurations using a bi-directional Mamba block (MetaSSM) across all experiments, as shown in Fig. 1(b). Initial experiments demonstrate that replacing the EncMetaBlock and MetaBlock modules with two sequential Mamba-based modules (creating a SwinUMamba model) or replacing all three meta blocks (forming a fully Mamba-based VM-UNet model) provides notable improvements of 1.9% and 1.8%, respectively. However, solely replacing DecMetaBlock and MetaBlock reduces performance to 89.4%, suggesting that placing Mamba modules within the encoder and bottleneck is optimal.

Further experiments indicate that a hybrid configuration—two convolutional layers followed by a Mamba module—achieves superior performance compared to either purely convolutional or purely Mamba-based modules. Fig. 2 (b) visualizes intensity distributions of feature maps extracted from a pre-trained model, confirming that feature maps within residual connections exhibit lower variance than those outside residual connections. Since SSMs struggle with high-variance inputs, embedding Mamba modules within residual connections proves beneficial.

Finally, we explore the optimal placement of Mamba modules within the hybrid model, comparing configurations with Mamba either inside or outside residual connections. Our experiments confirm that embedding Mamba within residual connections enhances segmentation accuracy by an additional 0.5%, thus guiding the design choice for MM-UNet.

Consequently, MM-UNet employs hybrid modules that integrate Mamba within residual connections in the encoder and bottleneck to achieve the highest segmentation performance.

### 3.4 SCAN DESIGN

In this subsection, we explore various scan order strategies for the Mamba block, named MetaSSM, as illustrated in Fig. 3. Given that medical datasets commonly contain 3D spatial information, the flattening of these 3D images into 1D sequences can follow multiple possible orders. For instance,

| Method | Spleen | R.Kd | L.Kd | GB | Eso. | Liver | Stom. | Aorta | IVC | Panc. | RAG | LAG | Duo. | Blad. | Pros. | Average |
|---|---|---|---|---|---|---|---|---|---|---|---|---|---|---|---|---|
| UNETR Hatamizadeh et al. (2022) | 0.928 | 0.913 | 0.903 | 0.719 | 0.763 | 0.955 | 0.849 | 0.922 | 0.838 | 0.766 | 0.663 | 0.663 | 0.662 | 0.815 | 0.744 | 0.807 |
| nnFormer Zhou et al. (2021) | 0.950 | 0.948 | 0.944 | 0.789 | 0.784 | 0.967 | 0.914 | 0.931 | 0.868 | 0.828 | 0.654 | 0.695 | 0.759 | 0.865 | 0.773 | 0.845 |
| SwinUNETR Hatamizadeh et al. (2021) | 0.954 | 0.954 | 0.950 | 0.819 | 0.852 | 0.972 | 0.919 | 0.955 | 0.911 | 0.875 | 0.775 | 0.801 | 0.816 | 0.895 | 0.812 | 0.884 |
| SwinUNETRv2 He et al. (2023) | 0.959 | 0.962 | 0.958 | 0.842 | 0.867 | 0.976 | 0.933 | 0.957 | 0.920 | 0.889 | 0.783 | 0.812 | 0.843 | 0.913 | 0.836 | 0.897 |
| 3D UX-Net Lee et al. (2022) | 0.955 | 0.956 | 0.953 | 0.826 | 0.858 | 0.972 | 0.922 | 0.955 | 0.915 | 0.881 | 0.781 | 0.809 | 0.820 | 0.902 | 0.823 | 0.889 |
| nn-UNet Isensee et al. (2019) | 0.951 | 0.961 | 0.956 | 0.826 | 0.869 | 0.973 | 0.931 | 0.957 | 0.923 | 0.880 | 0.784 | 0.809 | 0.846 | 0.898 | 0.827 | 0.893 |
| VMUNet Ruan & Xiang (2024) | 0.958 | 0.967 | 0.964 | 0.847 | 0.866 | 0.976 | 0.938 | 0.957 | 0.923 | 0.882 | 0.785 | 0.805 | 0.852 | 0.904 | 0.819 | 0.896 |
| SwinUMamba Liu et al. (2024a) | 0.958 | 0.967 | 0.965 | 0.846 | 0.870 | 0.976 | 0.940 | 0.958 | 0.922 | 0.883 | 0.783 | 0.811 | 0.856 | 0.902 | 0.818 | 0.897 |
| UMamba Ma et al. (2024) | 0.971 | 0.969 | **0.967** | 0.863 | 0.879 | **0.979** | 0.944 | 0.960 | 0.929 | 0.894 | 0.795 | 0.819 | 0.864 | 0.914 | 0.830 | 0.905 |
| MM-UNet (Ours) | **0.973** | **0.970** | 0.967 | **0.876** | **0.884** | 0.979 | 0.946 | 0.962 | 0.931 | 0.899 | 0.802 | 0.826 | 0.871 | 0.920 | 0.842 | **0.910** |

Table 2: Comparison of MM-UNet with state-of-the-art methods on the AMOS testing dataset, evaluated by Dice Score. For a fair comparison, all results are based on 5-fold cross-validation without any ensembles. The best results are indicated in **bold**.

a Depth-Height-Width (DHW) order flattens the image by scanning along width first, then height, and finally depth. Similarly, alternative orders (DWH, WDH, WHD, HDW, and HWD), each with their respective inverse scan order, can be employed, as shown in Fig. 3(2a)-(2l). Additional scanning patterns are illustrated in Fig. 3(2m)-(2p), including 2D, 3D window-based, zigzag, and inclined scans.

Our experiments, summarized in Table 1, investigate the effectiveness of these different scan orders. Beginning with the standard one-dimensional (1D) DHW scan (baseline model B1), we achieve a Dice score of 0.902. By introducing a complementary scan order in the opposite direction, forming the 1D BiScan (B2), performance improves by 0.008, resulting in a Dice score of 0.910. This improvement highlights the effectiveness of bi-directional scanning in mitigating prediction errors caused by discontinuities when flattening spatial dimensions into a single sequence.

We further extend our investigation by incorporating additional scan order pairs. However, adding a third independent scan direction (HWD) to the 1D BiScan does not yield improvements, indicating that unpaired scan orders introduce detrimental discontinuities. Similarly, extending the approach to three pairs (B4) or even six pairs (B5) of opposite-direction scan orders achieves Dice scores of 0.908 and 0.907, respectively, demonstrating that adding more pairs beyond one does not provide additional benefit and may introduce redundancy.

Additionally, we investigate other scan strategies, such as a two-dimensional (2D) scan (B6), which underperforms due to not adequately modeling temporal continuity. A 3D window-based scan method (B7), which divides the entire 3D volume into multiple non-overlapping windows, achieves comparable results (0.902) to the 1D scan. However, this method inherently increases discontinuities because of frequent jumps between window boundaries. The Zigzag scan (B8) achieves performance similar to the 1D BiScan (0.909), while the Inclined scan (B9) performs slightly worse (0.901) due to increased discontinuities.

These findings indicate that the optimal scan strategy for MetaSSM is to use a simple bi-directional approach, consisting of just one pair of opposite-direction orders. This balanced approach sufficiently captures spatial continuity and effectively mitigates prediction errors arising from discontinuities during the flattening process. Consequently, we adopt the 1D BiScan, comprising DHW and its inverse order, flip(DHW), as the standard scan strategy in our proposed MM-UNet.

## 4 EXPERIMENTS

### 4.1 DATASETS AND EVALUATION METRICS

We conduct experiments using two publicly available datasets: the AMOS22 Abdominal CT Organ Segmentation dataset Ji et al. (2022) and the Synapse challenge dataset Landman et al. (2015). **(i)** The AMOS22 dataset contains 300 abdominal CT scans with manual annotations for 16 anatomical structures, which serve as the basis for multi-organ segmentation tasks. The testing set comprises 200 images, and we evaluate our model using the AMOS22 leaderboard. **(ii)** The Synapse dataset includes 30 cases of abdominal CT scans. Following established split strategies Hatamizadeh et al. (2021), we use 24 cases for training and 4 cases for validation. Performance is assessed using the average Dice Similarity Coefficient (DSC) across 13 abdominal organs.

| Method | Spl. | R.Kd | L.Kd | GB | Eso. | Liv. | Stom. | Aorta | IVC | Veins | Panc. | AG | DSC |
|---|---|---|---|---|---|---|---|---|---|---|---|---|---|
| TransUNet Chen et al. (2021) | 0.952 | 0.927 | 0.929 | 0.662 | 0.757 | 0.969 | 0.889 | 0.920 | 0.833 | 0.791 | 0.775 | 0.637 | 0.838 |
| 3D UX-Net Lee et al. (2022) | 0.946 | 0.942 | 0.943 | 0.593 | 0.722 | 0.964 | 0.734 | 0.872 | 0.849 | 0.722 | 0.809 | 0.671 | 0.814 |
| UNETR Hatamizadeh et al. (2022) | 0.968 | 0.924 | 0.941 | 0.750 | 0.766 | 0.971 | 0.913 | 0.890 | 0.847 | 0.788 | 0.767 | 0.741 | 0.856 |
| Swin-UNETR Hatamizadeh et al. (2021) | **0.971** | 0.936 | 0.943 | **0.794** | 0.773 | **0.975** | 0.921 | 0.892 | 0.853 | **0.812** | 0.794 | **0.765** | 0.869 |
| nnUNet Isensee et al. (2019) | 0.942 | 0.894 | 0.910 | 0.704 | 0.723 | 0.948 | 0.824 | 0.877 | 0.782 | 0.720 | 0.680 | 0.616 | 0.802 |
| nnFormer Zhou et al. (2021) | 0.935 | 0.949 | 0.950 | 0.641 | 0.795 | 0.968 | 0.901 | 0.897 | 0.859 | 0.778 | 0.856 | 0.739 | 0.856 |
| VMUNet Ruan & Xiang (2024) | 0.962 | 0.948 | 0.951 | 0.581 | 0.784 | 0.968 | 0.866 | 0.900 | 0.861 | 0.754 | 0.815 | 0.722 | 0.843 |
| SwinUMamba Liu et al. (2024a) | 0.961 | 0.950 | 0.951 | 0.614 | 0.777 | 0.970 | 0.876 | 0.904 | 0.866 | 0.755 | 0.822 | 0.712 | 0.847 |
| UMamba Ma et al. (2024) | 0.962 | 0.950 | 0.950 | 0.627 | 0.789 | 0.970 | 0.888 | 0.909 | 0.871 | 0.769 | 0.835 | 0.714 | 0.853 |
| MM-UNet (Ours) | 0.967 | **0.952** | **0.953** | 0.660 | **0.806** | 0.973 | **0.932** | **0.916** | **0.886** | 0.803 | **0.862** | 0.740 | **0.871** |

Table 3: Comparison of MM-UNet with state-of-the-art methods on Synapse dataset (DSC in %). The best results are highlighted in bold.

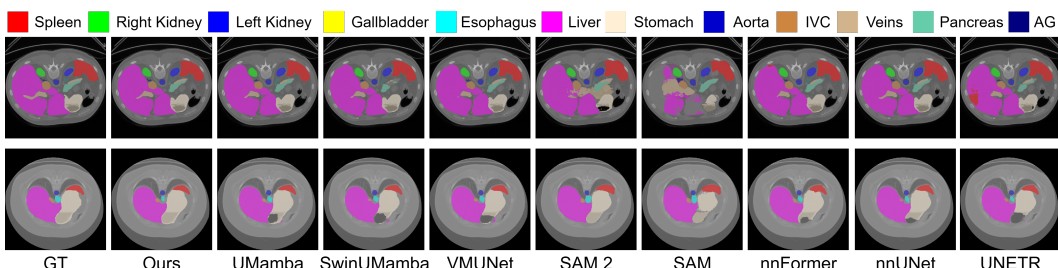

Figure 6: Qualitative comparison on Synapse dataset. MM-UNet is the most precise for each class.

## 4.2 IMPLEMENTATION DETAILS

We adopt the nnUNet framework for training, modifying only the network architecture while keeping all other configurations consistent. Data augmentation strategies follow those used in nnUNet. The initial learning rate is set to 0.001, and we apply a polynomial decay strategy as defined in Eq. equation 5:

$$lr(e) = init\_lr \times \left( 1 - \frac{e}{\text{MAX\_EPOCH}} \right)^{0.9}, \tag{5}$$

where $e$ denotes the current epoch, and MAX_EPOCH is set to 1000, with each epoch consisting of 250 iterations. We use SGD as the optimizer with a momentum of 0.99. The weight decay is set to $3 \times 10^{-5}$. The batch size is set to 2 for all experiments. The loss function is a combination of cross-entropy loss and Dice loss. We employ a 5-fold cross-validation strategy for results presented in Tab. 2.

**Deep Supervision.** Our network is trained with deep supervision. Auxiliary losses are applied in the decoder at the last three stages (corresponding to the three largest resolutions). For each deep supervision output, the ground truth segmentation mask is downsampled to match the corresponding resolution. The final training objective is computed as:

$$\mathcal{L} = w_1 \cdot \mathcal{L}_1 + w_2 \cdot \mathcal{L}_2 + w_3 \cdot \mathcal{L}_3, \tag{6}$$

where the weights decrease by half with each reduction in resolution (*i.e.,* $w_2 = \frac{1}{2}w_1$, $w_3 = \frac{1}{4}w_1$). The weights are normalized to sum to 1. Additionally, the resolution of $\mathcal{L}_1$ is twice that of $\mathcal{L}_2$ and four times that of $\mathcal{L}_3$.

## 4.3 COMPARISON WITH STATE-OF-THE-ART METHODS

**Results on AMOS 2022 Dataset.** We present the quantitative results of our experiments on the AMOS 2022 dataset in Tab. 2, comparing our proposed MM-UNet against widely used segmentation methods. These include convolution-based methods (nnUNet Isensee et al. (2019); Lee et al. (2022)), transformer-based methods (UNETR Hatamizadeh et al. (2022), SwinUNETR Hatamizadeh et al. (2021), and nnFormer Zhou et al. (2021)), and Mamba-based methods (VMUNet Ruan & Xiang (2024), SwinUMamba Liu et al. (2024a), and UMamba Ma et al. (2024)). For a fair comparison, all methods undergo 5-fold cross-validation without ensembling.

Our MM-UNet outperforms all existing methods on most organs, achieving state-of-the-art performance in DSC. Specifically, it surpasses nnUNet and 3D UX-Net by 1.7% and 2.1% in DSC,

respectively. Meanwhile, our method is better than all Mamba-based methods. Given the complexity of the AMOS 2022 dataset, these results demonstrate the effectiveness of our proposed method.

**Results on Synapse Dataset.** We present the quantitative results of our experiments on the Synapse dataset in Table 3, comparing our proposed MM-UNet against several leading convolution-based methods (VNet Ronneberger et al. (2015), nnUNet Isensee et al. (2019)), transformer-based methods (TransUNet Chen et al. (2021), SwinUNet Cao et al. (2021), nnFormer Zhou et al. (2021)), and Mamba-based methods (VMUNet Ruan & Xiang (2024), SwinUMamba Liu et al. (2024a), and UMamba Ma et al. (2024)).

Our MM-UNet achieves a new state-of-the-art performance, outperforming all existing methods. Specifically, it surpasses nnFormer and nnUNet by 1.5% and 6.9% in DSC, respectively, on this highly competitive dataset. Notably, our model excels in segmenting large organs such as the liver and spleen, which benefit from the ability of SSMs to capture long-range dependencies.

Fig. 6 presents qualitative comparisons against representative methods, demonstrating that MM-UNet generates more accurate segmentations, particularly for the liver and spleen. These results confirm the robustness and effectiveness of our proposed approach.

### 4.4 ATTENTION MAPS FOR MAMBA

Inspired by VMamba Liu et al. (2024b), to gain an intuitive and in-depth understanding of MetaSSM, we further visualize the attention values in $QK^T$ illustrated in Fig. 4. Each attention map effectively captures image patterns across the temporal dimension, even when the 3D medical images are flattened into a 1D sequence as input for the MetaSSM blocks. Unlike attention layers that rely on a large number of parameters to establish relationships at the pixel level, SSMs achieve superior attention maps using only a few parameters. These remarkable characteristics highlight the motivation for using SSMs in medical image segmentation.

## 5 CONCLUSION

In this paper, we propose MM-UNet, a unified U-shaped encoder-decoder architecture with skip connections, designed specifically for medical image segmentation. MM-UNet is the first unified framework capable of integrating and representing existing Mamba-based models while leveraging the strengths of both convolutional neural networks (CNNs) and state space models (SSMs). Our work addresses key challenges in adapting SSMs to medical images, including their difficulty in handling high-variance data and their inability to model discontinuities that arise from flattening multi-dimensional images into a one-dimensional sequence.

Through extensive exploration of SSM properties, we determine an optimal macro design for MM-UNet, ensuring that SSM-based modules are placed where they contribute most effectively. We also propose a novel scan order strategy for processing 3D medical images, leveraging bi-directional scanning to mitigate the impact of discontinuities and improve segmentation accuracy.

To validate our approach, we conduct comprehensive experiments on three challenging medical segmentation datasets: AMOS Ji et al. (2022) and Synapse Landman et al. (2015). The results demonstrate that MM-UNet achieves state-of-the-art performance across multiple benchmarks. Specifically, our model achieves a Dice score of 91.00% on the AMOS2022 dataset Ji et al. (2022), surpassing nnUNet by 1.7%. Furthermore, MM-UNet achieves state-of-the-art performance on the Synapse dataset, a Dice score of 87.1%, reinforcing its robustness and generalization capability.

Beyond achieving competitive segmentation accuracy, our findings provide deeper insights into the structural properties of SSMs in the context of medical image segmentation. By systematically investigating how SSMs interact with CNN-based architectures and proposing an effective hybrid design, we bridge the gap between traditional deep learning models and emerging sequence-based approaches in the vision domain. Future work will explore further optimizations in integrating SSMs within medical image segmentation frameworks, particularly adapting MM-UNet to additional modalities such as MRI and ultrasound. Additionally, we plan to extend MM-UNet to broader medical imaging tasks beyond segmentation, including registration and classification, to further demonstrate its versatility and impact in biomedical image analysis.

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
