# OpenReview forum: "MM-UNet: Meta Mamba UNet for Medical Image Segmentation"
_ICLR.cc/2026/Conference — ICLR 2026 Conference Withdrawn Submission_

### Official Review · Reviewer_QW7b · 2025-10-28

**Soundness:** 2
**Presentation:** 2
**Contribution:** 2
**Rating:** 4
**Confidence:** 3

**Summary:**

This paper introduces MM-UNet, a unified U-shaped encoder-decoder architecture designed to adapt State Space Models (SSMs) for volumetric medical image segmentation. The authors identify key challenges in applying SSMs to medical imaging, including handling high-variance data and spatial discontinuities from flattening 3D images. Through experiments with replaceable meta-blocks, the paper determines that a hybrid design, placing Mamba modules after convolutional layers within residual connections, yields optimal performance. Extensive validation on AMOS22 and Synapse datasets shows good results,

**Strengths:**

1. MM-UNet serves as a flexible framework capable of representing various Mamba-based models (e.g., VM-UNet, U-Mamba), enabling systematic comparison of architectural choices.

2. Experiments validate the hybrid residual design, showing that embedding SSMs within residual connections reduces feature variance and improves segmentation accuracy.

3. Achieves good results on two challenging benchmarks,

**Weaknesses:**

1. The work primarily optimizes existing SSM components (e.g., scan orders, residual connections) without advancing the theoretical foundations of SSMs (e.g., novel state matrix designs or better long-range modeling mechanisms)

2.  Experiments are limited to abdominal CT datasets (AMOS22, Synapse). Generalizability to other modalities (e.g., MRI) or anatomical regions (e.g., brain) remains unverified.

3. The use of nnUNet’s preprocessing and training protocols may inflate performance gains attributable to the architecture itself. Ablations on data augmentation or preprocessing strategies would clarify MM-UNet’s standalone contribution.

4. Figure 6 provides basic segmentation visuals but lacks examples of edge cases, small organs, or failure modes. A deeper analysis of attention mechanisms (e.g., how Mamba’s input-dependent selection operates on medical features) would strengthen the study.

**Questions:**

see the weakness

---

### Official Review · Reviewer_oe6u · 2025-11-01

**Soundness:** 2
**Presentation:** 2
**Contribution:** 2
**Rating:** 2
**Confidence:** 5

**Summary:**

This paper proposes a unified U-shaped encoder–decoder architecture, Meta Mamba UNet (MM-UNet), which aims to fully exploit the advantages of State Space Models (SSMs) while mitigating their limitations. Specifically, MM-UNet integrates a hybrid module that embeds SSMs into residual connections to reduce variance and enhance performance.
The proposed MM-UNet achieves state-of-the-art performance on two public medical image segmentation datasets, AMOS2022 and Synapse.

**Strengths:**

1. The authors identify and address two key challenges of applying SSMs to high-resolution medical images.

2. Despite several presentation issues, the manuscript is relatively well-organized.

**Weaknesses:**

1. Although the experiments show that SSM-based architectures outperform CNN- and ViT-based counterparts, the paper lacks a reasonable explanation of why SSMs perform better. In particular, what types of information are SSMs capturing that lead to superior performance?

2. The application of SSMs to medical image segmentation has already been extensively studied [1,2]. While MM-UNet achieves strong results, its contribution is incremental. Furthermore, both the proposed residual design and bidirectional scanning can already be found in existing SSM literature [3,4]; thus, these techniques are not novel.

3. The authors state in line 409 that five-fold cross-validation was used, yet the reported results do not include statistical measures such as mean and standard deviation.

4. The performance improvement over recent works is marginal. For example, in Table 2, MM-UNet improves over UMamba by only 0.005, and in Table 3, it improves over Swin-UNet by merely 0.002.

5. A recent study [5] has provided an in-depth discussion on the fairness and rigor of comparisons in existing medical image segmentation research. How do the authors address the issue of fairness in the comparative evaluation of MM-UNet?

[1] Vm-unet: Vision mamba unet for medical image segmentation, ACM Transactions on Multimedia Computing, 2024

[2] SegMamba: Long-Range Sequential Modeling Mamba for 3D Medical Image Segmentation, MICCAI, 2024

[3] VMamba: Visual State Space Model, NeurIPS’24 [2] Vision mamba: Efficient visual representation learning with bidirectional state space model, ICML, 2024

[4] GroupMamba: Efficient Group-Based Visual State Space Model, CVPR, 2025

[5] nnU-Net Revisited: A Call for Rigorous Validation in 3D Medical Image Segmentation, MICCAI, 2024.

**Questions:**

1. There are numerous citation formatting problems. Specifically, the authors should pay close attention to the proper use of \citep and \citet in LaTeX.

2. Several symbols are not clearly explained, including init_lr in Equation (5) and L1, L2, L3 in Equation (6).

3. The authors inconsistently use both “nnUNet” and “nn-UNet”; the terminology should be unified throughout the manuscript.

4. In Figure 7, the visual differences between segmentation results of different methods are barely discernible.

---

### Official Review · Reviewer_iVKL · 2025-11-01

**Soundness:** 3
**Presentation:** 2
**Contribution:** 2
**Rating:** 2
**Confidence:** 3

**Summary:**

This paper presents MM-UNet (Meta Mamba UNet), a U-shaped encoder–decoder framework that integrates State Space Models (SSMs)—specifically Mamba blocks—into a UNet-like architecture for 3D medical image segmentation. The authors claim that SSMs can model long-range dependencies with linear computational complexity, offering a more efficient alternative to Transformers. They further introduce a “meta-block” design that unifies CNN and Mamba components, allowing flexible hybrid configurations, and a bi-directional scan strategy to mitigate discontinuities when flattening 3D volumes. Experiments on AMOS22 and Synapse datasets demonstrate competitive Dice scores (91.0 % and 87.1 %, respectively), surpassing several CNN- and Mamba-based baselines.

**Strengths:**

1. The meta-block design elegantly unifies CNN-only, Mamba-only, and hybrid configurations within a single architecture.
2. Results on two benchmarks include both quantitative and qualitative analyses, clearly showing how different block choices affect performance.
3. The paper is well written, logically structured, and visually clear。

**Weaknesses:**

1. The method mainly combines existing ideas from prior Mamba-UNet variants (e.g., U-Mamba, VM-UNet) without introducing new learning principles or theoretical insights.
2. Despite emphasizing SSMs’ linear complexity, the paper omits runtime, FLOPs, memory usage, and parameter count comparisons. The claimed computational benefits remain unsubstantiated.
3. Experiments are restricted to CT, whereas the title and framing suggest general “volumetric medical image segmentation.” Validation on additional modalities (e.g., MRI from AMOS) would strengthen generalizability claims. The improvements are also not consistent: Gains vary irregularly across organs (e.g., Gallbladder +5 % on AMOS22 vs –4 % on Synapse), suggesting limited robustness.
4. Foundation-model-based approaches (e.g., finetuned SAM, Med-SAM, SAM 2) are omitted, even though they represent current strong baselines in volumetric segmentation.

**Questions:**

1. Can the authors provide quantitative comparisons of computational cost (training/inference time, FLOPs, memory, parameters) versus nnUNet and U-Mamba?
2. How does MM-UNet perform against finetuned SAM or Med-SAM models? Including these baselines would contextualize performance against current state-of-the-art.
3. Can authors test on some non-CT datasets to show the method's robustness against modality?

---

### Official Review · Reviewer_WREn · 2025-11-03

**Soundness:** 3
**Presentation:** 3
**Contribution:** 3
**Rating:** 6
**Confidence:** 3

**Summary:**

The paper proposes MM-UNet, a U-shaped encoder–decoder for 3D medical image segmentation that “unifies” prior Mamba-based variants via meta-blocks that can be CNN, SSM (Mamba), or hybrid. The main empirical finding is that a hybrid block with two conv layers followed by a Mamba block placed inside residual connections in the encoder and bottleneck performs best. The authors also argue SSMs struggle with high-variance inputs and discontinuities introduced by flattening 3D volumes; they mitigate this with (i) residual placement to reduce feature variance and (ii) a bi-directional scan order (DHW and its reverse) for the Mamba scans. On AMOS22 and Synapse, MM-UNet reports SOTA Dice (AMOS22: 91.0%; Synapse: 87.1%) without ensembling. Figures 1–5 and Tables 1–3 detail the architecture, ablations (scan orders), and quantitative results (notably Table 2/3).

**Strengths:**

•	Practical architectural insight. The paper pinpoints where Mamba helps in a U-shape and shows inside-residual placement in the encoder/bottleneck outperforms alternatives (Fig. 1b, page 2).
	•	Simple scan strategy that works. The bi-directional scan (DHW + reverse) balances continuity vs. discontinuity and outperforms more elaborate variants (Table 1, page 4).
	•	Strong empirical results. Consistent gains on AMOS22 (91.0% Dice) and Synapse (87.1% Dice) with per-organ improvements (Tables 2–3, pages 7–8) and qualitative examples (Fig. 6, page 8).
	•	Clear motivation for SSM idiosyncrasies. The paper articulates why 1D scans of 3D volumes create discontinuities and why Mamba may be sensitive to high-variance inputs; Fig. 2 (page 3) and Fig. 4 (page 5) give intuition.

**Weaknesses:**

•	Limited statistical rigor. Improvements are mostly reported as mean Dice without variance/CI across folds/cases. Given the margins (~0.5–1.0% in places), significance tests or per-case distributions would strengthen claims (Tables 2–3).
	•	Compute/efficiency not characterized. There is no report of FLOPs/params/latency/VRAM vs. nnUNet, UMamba, VM-UNet, etc. Since SSM placement may affect memory/computation, a fair efficiency comparison is important.
	•	Ablations stop short of causality. The “SSMs struggle with high-variance inputs” finding (Fig. 2b) is compelling but anecdotal; more thorough variance measurements (e.g., across layers/datasets/training stages) or controlled noise/contrast studies would solidify the claim.
	•	Fairness of test-time protocol. The inference procedure uses 8 flips across axial/coronal/sagittal + 3D Gaussian weighting (Fig. 5, page 5–6). It’s unclear if competitors in Tables 2–3 used identical TTA; even if “no ensembles” were used, TTA can materially boost Dice.
	•	Scope of generalization. Experiments are confined to CT abdomen; the method likely generalizes, but MRI/ultrasound or additional CT datasets would bolster external validity.
	•	“Unified” positioning may be overstated. The meta-block framework is a clean abstraction, but prior works already vary where/how to insert Mamba. Emphasizing the integration insights (residual placement, scan order) rather than a “first unified framework” claim would be more precise.

**Questions:**

1.	Significance & variance: Can you report mean±std Dice across folds and per-case distributions on AMOS22/Synapse? Are the gains statistically significant (e.g., Wilcoxon test)?
	2.	Compute profile: Please provide params, FLOPs, peak VRAM, and inference latency for MM-UNet vs. nnUNet, UMamba, VM-UNet at the same input size.
	3.	TTA fairness: Did you re-run baselines with the same 8-flip + Gaussian weighting TTA? If not, how much of your gain remains when you disable TTA for MM-UNet?
	4.	Residual placement rationale: Beyond Fig. 2b, can you quantify feature variance empirically across layers and show how residual placement reduces it (e.g., histograms/statistics over the training set)?
	5.	Ablations on decoder: You conclude encoder/bottleneck Mamba is best. Could you share decoder-only Mamba results with the same TTA/training to isolate effects?
	6.	Scan order generality: Does the bi-directional scan remain optimal for other volume shapes (anisotropic spacing) or modalities? Any results on MRI or different CT organs?
	7.	Robustness: How does performance change under intensity shifts/noise (e.g., simulating different scanners or protocols)?
	8.	Release plan: Will code/configs be released to reproduce the nnUNet-compatible training and the exact inference pipeline?

---

### Note · Authors · 2025-11-14

I have read and agree with the venue's withdrawal policy on behalf of myself and my co-authors.